# Characterization of Magnetic Nanoparticles from the Shells of Freshwater Mussel *L. fortunei* and Marine Mussel *P. perna*

Antonio Valadão Cardoso [1,*](ID), Clara Carvalho Souza [2], Maria Sylvia Dantas [3], Camila Schults Machado [2], Erico Tadeu Freitas [4], Alisson Krohling [5], Veronica Martins Rosario [2], Giancarlo Ubaldo Nappi [6] and Luiz Dias Heneine [6]

[1] School of Design, State University of Minas Gerais (UEMG), Belo Horizonte 30140-092, MG, Brazil
[2] Center for Bioengineering of Invasive Species (CBEIH), Belo Horizonte 31035-536, MG, Brazil
[3] Metallurgy Department, School of Engineering (UFMG), Belo Horizonte 31270-901, MG, Brazil; dantas.sylvia@gmail.com
[4] Materials Science and Engineering, ACMAL, Michigan Technological University (MTU), Houghton, MI 49931, USA
[5] Center for Nuclear Technology Development (CDTN), Belo Horizonte 31270-901, MG, Brazil
[6] Laboratory of Applied Immunology, E Dias Foundation (FUNED), Belo Horizonte 30510-010, MG, Brazil; giancarlo.nappi@funed.mg.gov.br (G.U.N.); heneinel@gmail.com (L.D.H.)
* Correspondence: antonio.cardoso@uemg.br

**Abstract:** Magnetite ($Fe_3O_4$) nanoparticles were extracted from the shells of freshwater *Limnoperna fortunei* (Dunker 1857) and marine *Perna perna* (Linnaeus 1758) mussels, followed by full physical and chemical characterization using ICP-OES, UV–Vis, EDX, Raman, and XRD spectroscopy, VSM magnetometry, and SEM and TEM techniques. Considering their spatial distribution, the ferrimagnetic particles in the shells had low concentration and presented superparamagnetic behavior characteristics of materials of nanometric size. Transmission electron microscopy (TEM, especially HRTEM) indicated round magnetic particles around 100 nm in size, which were found to be aggregates of nanoparticles about 5 nm in size. The TEM data indicated no iron oxide particles at the periostracum layer. Nevertheless, roughly round iron (hydr)oxide nanoparticle aggregates were found in the nacre, namely, the aragonite layer. As the aragonite layer is responsible for more than 97% of the shell of *L. fortunei* and considering the estimated size of the magnetic nanoparticles, we infer that these particles may be distributed homogeneously throughout the shell.

**Keywords:** *Limnoperna fortunei*; nanoparticles; magnetism; shell; freshwater bivalve; *Perna perna*; marine bivalve

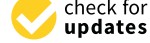



## 1. Introduction

The first and only reported observation of the presence of magnetite ($FeO.Fe_2O_3$) in mollusks was in the radula of chitons in 1962 [1]. Since then, the number of studies describing the presence of biomineralized magnetite in animal species has grown steadily [2,3]; however, to the best of our knowledge, no studies have reported the presence of iron (hydr)oxides in other mollusks to date, apart from chitons and limpets.

The motivation for this present work began while performing another experiment to produce chitosan from *Limnoperna fortunei* (Dunker 1857) shells in our laboratory. During the shell demineralization process, we observed the presence of black particle aggregates attached to the magnetic stirring bar (PTFE-coated; Figures 1a and 2a). This was an intriguing finding. After repeating the same protocol many times, we found that the extremely small magnetic particles were derived from the shell.

While checking the literature for any clue, we noticed that some scientific papers have reported the presence of hematite ($\alpha$-$Fe_2O_3$) in the shells of bivalves [4–6]. The hematite content in the shell composition in these studies was reported to be much lower than

1 wt%. As the researchers aimed to facilitate the use of shells as raw material for plastics, catalysts, and so on in these studies, they calcined the shell close to 1000 °C. In doing so, the magnetite and any other existing iron compounds would have been oxidized and transformed into the more stable end-phase hematite ($\alpha$-Fe$_2$O$_3$) [7].

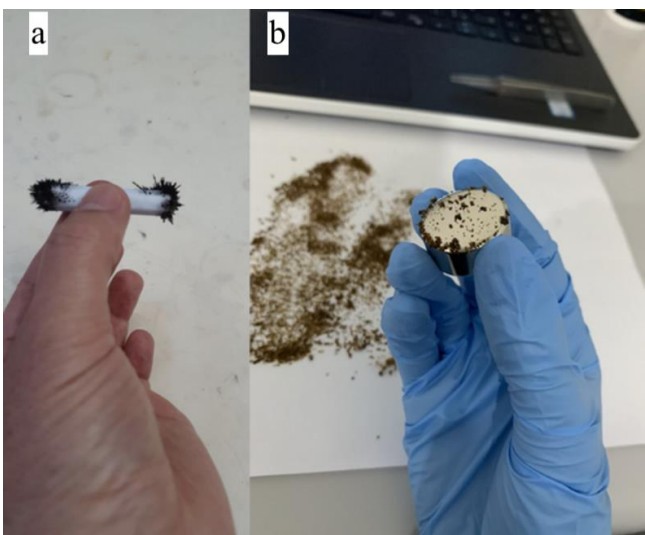

**Figure 1.** Images of the magnetic particles extracted from the demineralized shells of (**a**) freshwater bivalve *Limnoperna fortunei* and (**b**) marine bivalve *Perna perna*. The extracted particles are attached to a magnetic (PTFE-coated) stirrer bar (**a**) and neodymium (NdFeB) magnet (**b**).

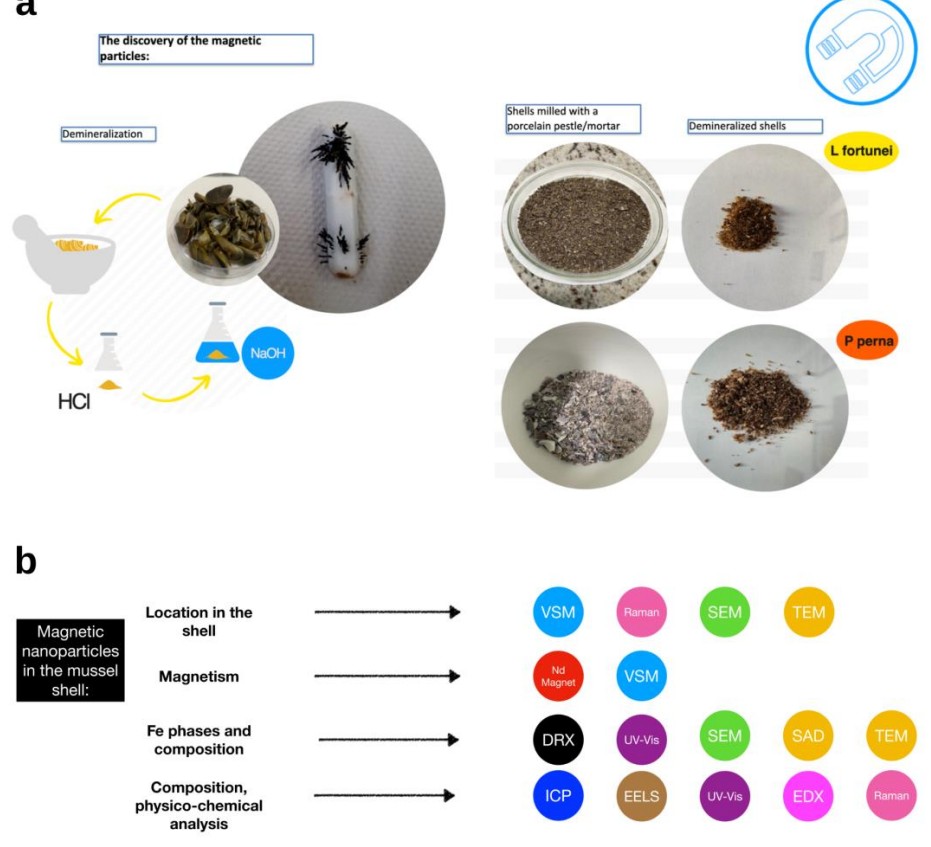

**Figure 2.** (**a**) Magnetic particles observed during demineralization of mussel shells; and (**b**) experimental setup after discovery of magnetic nanoparticles in mussel shells.

Ferrimagnetic magnetite ($FeO.Fe_2O_3$) [8,9] is an iron ore. Its crystallographic phases are some of the most interesting among iron oxides, especially in terms of their nanoscale occurrence. Iron oxide exhibits three different crystalline polymorphs with unique magnetic properties. When heating magnetite, it first transforms into maghemite ($\gamma$-$Fe_2O_3$) at around 300 °C, then to hematite ($\alpha$-$Fe_2O_3$) at around 450 °C. Note that these values are for bulk material; as the surface-to-volume ratio plays a role in thermodynamic processes, the oxidation of magnetite to maghemite at the nanoscale could occur at temperatures close to ambient, as well as inside living matter. As a matter of fact, Petit et al. [10] observed the presence of iron-containing micro/nanospheres in the epithelial cells of the mussel mantle.

Regarding our previous observation in the laboratory, we came up with some questions and hoped to find clues to help us address them: Have the magnetic nanoparticles present in the bivalve shells simply been ingested or absorbed by the gills of these suspension feeders? Have the magnetic iron oxide nanoparticles nucleated and formed into their bodies? More intriguingly, do the magnetic iron oxide phases play a role in shell biomineralization? Of course, we did not arrive at answers to all of these questions, but we believe that our findings reported here may lead to such knowledge. Understanding shell biomineralization and whether (and how) iron oxide species could potentially play a role in this process is a fascinating topic. It is well-known that iron cations change the crystal growth of calcite and aragonite, which are both components of the bivalve shells. Di Lorenzo et al. [11], for example, have shown that aqueous $Fe^{2+}$ stabilizes aragonite at the expense of calcite. Studies on the biomineralization of bivalve shells seldom address the presence, influence, or role of iron in the process of shell formation; rather, they tend to focus on the source of calcium.

Historically, experimental work (1953) has revealed that one valve–mantle set, severed from the bivalve body and kept in seawater for in vitro studies, continued to biomineralize [12] for several days. Being an intact system with respect to the relationship between the mantle, the extra-pallial space (ESP), and the shell, it is clear that the calcium was not derived from metabolism. More than seventy years ago, another experimental work [13] using valve–mantle preparations with the radioisotope $^{45}$Ca revealed that the calcium present in the mussel mantle remained there and was not supplying the shell biomineralization process. Only in this century have advances regarding the fundamental role of hemocytes in biomineralization been acquainted and established [14]. The sheet of outer epithelial cells of the mantle facing the shell forms the frontier where the $CaCO_3$ nano-agglomerates (amorphous or crystalline) arrive at the extra-pallial site—the fluid-filled space where $CaCO_3$ biomineralization takes place.

To continue our investigation on the presence of magnetic particles in the mussel shells (see Experimental Setup in Figure 2b), we decided to utilize the same HCl-based demineralization protocol and search for magnetic material in the shell of the seawater bivalve *Perna perna* (Linnaeus 1758), also known as a brown mussel. As shown in Figure 1b, we also observed the presence of magnetic particles (henceforth, magnetic powder) attached to a neodymium (NdFeB) magnet after gently moving it above the demineralized shell powder (see the video in Supplementary Materials). Furthermore, we decided to demineralize the shells using ethylenediaminetetraacetic acid (EDTA), which is a well-known effective quelling organic acid for iron. The aim was to verify the presence of the Fe–EDTA complex using UV–Vis spectroscopy.

In our opinion, the occurrence of magnetic particles in the shell may turn out to be specific to *L. fortunei* and *P. perna* bivalves. Furthermore, it may open new opportunities for investigation of the role of magnetic nanoparticles in the assembly of hierarchical biomaterials.

## 2. Materials and Methods

### 2.1. Specimen Collection and Preparation

Specimens of *L. fortunei* were collected at the reservoir of the hydroelectric dam of Volta Grande (latitude 21°46′ South, longitude 42°32′ West), Minas Gerais, Brazil, and kept in an aquarium in our laboratory. Specimens of *P. perna* were acquired from the local market,

which came from mussel farms in the Santa Catarina Island area (located between latitudes 27°22′ and 27°50′ South and longitudes 48°25′ and 48°35′ West), Santa Catarina, Brazil. In the laboratory, *P. perna* specimens were kept in the freezer until the demineralization procedure.

## 2.2. Cleaning, Crushing, Sanding, and Powdering Shell Samples

After detaching from the mussel's body, the shells were first washed in a laboratory sink with running water until no trace of organic or inorganic materials (e.g., clays, sand, and other minerals) was noticed. The demineralization process was conducted using two different methodologies, in order to verify the presence of magnetic and iron particles as components of the shells. The golden mussel (*L. fortunei*) shells went through a cleaning process by submersion in a solution of sodium hypochlorite 12% (*w*/*v*), in order to remove all organic contents and other specimens that could be attached to the shells. The shells were incubated at ambient temperature for 5 min, followed by thorough washing with distilled water and drying in an oven at 70 °C. The dried samples were ground into a fine powder using an analytical mill (IKA A11 basic). This process was the same for both demineralization methodologies used.

## 2.3. Demineralization with Hydrochloric Acid

The demineralization methodology was adapted from [15]. A 1M hydrochloric acid solution was used in a proportion of 40 mL for each 1 g of shell powder. A total of 250 g of powdered shells was added slowly to the acid solution, under constant agitation using a magnetic stirrer. The association of the shell carbonate composition with the acid solution led to the formation of bubbles (in this case, carbon dioxide), which is evidence of demineralization. After adding all the powder, the mixture was left under constant stirring at room temperature for 6 h. Subsequently, it was centrifuged at $10,000\times g$ for 15 min at 25 °C using a Sorvall tabletop centrifuge (model St16-R-Thermo Scientific, Waltham, MA, USA). The supernatant was discarded, and a new solution of 1 M hydrochloric acid was added, maintaining the proportion of 40 mL acid to 1 g of shell, followed by incubation overnight at ambient temperature. The shell powder was washed with ultra-pure water in repeated centrifugation steps (as above) until the pH neutrality of the supernatant was observed. After the process, dark-colored particles were observed to have adhered to the magnetic bar. These particles were removed from the magnetic bar and washed with distilled water, followed by pure ethanol, and dried in an oven at 37 °C and then weighed.

## 2.4. Testing Magnetite Particles Dissolved in 1 M HCl Solution

Two experiments were performed: one with micrometer-sized and another with nanometer-sized magnetic particles. In the first experiment, 100 mg of natural magnetite ($FeO.Fe_2O_3$), from Turmalina (Latitude: 17°16′59″ South, Longitude: 42°44′7″ West), Minas Gerais, Brazil, with granulometry ranging between 200 and 300 μm were weighed using an electronic analytical balance (Mettler Toledo, model ME 204/A, Columbus, OH, USA). After weighing, the magnetite sample was immersed in 1 M hydrochloric acid (Sigma-Aldrich, Spruce St, St. Louis, MO, USA) solution for 24 h at 20–25 °C. After 24 h in acid, the magnetite particles were dried in air for many hours at ambient temperature. Finally, the dried sample was carefully collected and then weighed using the same balance. The weighing results were recorded.

In a second dissolution test, 100 mg of 30 nm diameter synthetic magnetite particles were immersed in 1 M hydrochloric acid solution for 24 h at 20–25 °C. After following the same protocol mentioned above, the final weight was recorded.

## 2.5. Demineralization with Ethylenediaminetetraacetic Acid (EDTA)

The use of a second process was adapted [16] from a methodology based on the use of EDTA (Sigma-Aldrich) to demineralize the shell while preserving the organic matrix present in it. A 0.014 g $L^{-1}$ acid–EDTA solution was prepared through the gradual addition of acid

to 100 mL of ultra-pure (Milli-Q) water under constant and vigorous stirring. Furthermore, in order to increase the solubility of the reagent, the temperature was increased to 50 °C (Solution A). By the end of the process, the pH of the solution was about 3.2, which is ideal for stable binding between EDTA and the iron [17] present in the shell. Then, 0.4 mg of the *L. fortunei* shell powder was added (sample 1) to 10 mL of the initial solution (Solution A). The solution was kept under continuous stirring for 5 days. We observed that the pH was a bit low by the end of the fifth day, closer to 2.0. Therefore, Na-EDTA was added to the solution (0.03 mol/L) for 2 days. By the seventh day, the sample was heated to 60 °C until it had a yellowish color, suggesting the presence of iron in the solution.

### 2.6. Direct Obtention of Magnetic Particles without Demineralization

Shells of the bivalves *L. fortunei* and *P. perna* were washed, rinsed, and thoroughly cleaned before being formed into powder material using common SiC sandpaper. Then, the powdered shell sample was placed on a paper sheet and scrutinized (from below) with a NdFeB commercial magnet to extract magnetic particles from the powdered shell (see Supplementary Materials and Raman experiments). The magnetic particles were collected for further experiments and characterization.

### 2.7. Characterization of the Magnetic Powder

The chemical composition of the *Limnoperna fortunei* and *Perna perna* shells was determined by induced coupled plasma–optical emission spectroscopy (ICP-OES, Perkin Elmer DV8300, Waltham, MA, USA). The CVAAS technique (Cold Vapor Atomic Absorption Spectrometry) was used to determine the Hg content. Results expressed as < (value) refer to the quantification limits of the technique. The used methodology was validated according to procedures recommended by official bodies (clean room class ISO-7 and laminar flow island class ISO-5).

The spectrophotometric measurements were carried out using a UV–VIS spectrophotometer (ThermoFisher model Biomate 160, Columbus, OH, USA) equipped with a Peltier thermostatted cell holder, 10 mm path-length quartz cuvettes, and a dual silicon photodetector (wavelength range 190–1100 nm). Experiments were carried out with a 1 nm wavelength step and two types of solutions were tested: (a) demineralized *L. fortunei* shell suspensions obtained using aqueous solutions of pure EDTA for demineralization, and (b) aqueous solutions with pure EDTA.

Energy-dispersive X-ray fluorescence spectroscopy (EDX, Shimadzu equipment, model EDX 7000, Kyoto, Japan) analysis was also performed. Approximately 5 g of *L. fortunei* shell and 5 g of *P. perna* shell were used for analysis of the concentration of atomic elements (in ppm). Previously, the shells of *L. fortunei* and *P. perna* were carefully washed with deionized water and brushed and dried at ambient temperature. The results of the EDX experiments are presented in the Supplementary Materials (Table S1 and EDX Spectra).

The concentrations of iron phases and other metals were assessed by X-ray diffraction (XRD). A diffractometer (model Empyrean, Panalytical, Westborough, MA, USA) equipped with a copper tube and a two-dimensional PIXEL 2X2 detector was used. The measurement was performed using CuKalpha1 radiation monochromatized with a hybrid double-mirror monochromator. The XRD measurements were performed in reflection mode, with the sample rotation period equal to 4 s. The experimental parameters for the measurements were 45 kV tension, 40 mA cathode current, and a 4–140 degree angle range. Quantitative composition analysis was performed using the Rietveld method and the quantification of phases was carried out using the Inorganic Crystal Structure Database (ICSD). Shells of *L. fortunei* and *P. perna* were carefully washed with deionized water and a brush, then dried at ambient temperature. Tests were carried out on the powdered shell, which was ground with a porcelain pestle and mortar.

A Vibrating Sample Magnetometer (VSM, LakeShore Model 7400, Westerville, OH, USA) was used to measure the magnetic properties of the magnetic powder. Hysteresis measurements were performed with an average of 10 s per point. The smallest gap (7.5 mm)

for the poles of the electromagnet available for the Kel-F polychlorotrifluoroethylene (PCTFE) sample holder was used. A reference measurement of the Kel-F sample holder was performed under the same conditions, and the sample holder signal was subtracted from the hysteresis curves of the shells. The Kel-F sample holder was ultrasonically cleaned in a neutral detergent and isopropyl alcohol before each measurement in order to remove any contaminants present. In addition, all sample handling—from sample mass quantification to placement in the VSM—was carried out with nitrile gloves and non-magnetic tweezers. We also carried out the process of canceling the magnetic remanence of the magnetometer before each hysteresis curve was obtained. The *P. perna* shells were broken into chunks using a pestle and mortar porcelain. The M (magnetization) × H (magnetic field) measurements were also normalized by the respective masses of the shell fragments, denoted as s1, s2, . . . , s10. In a third VSM experiment, powdered *L. fortunei* shell samples were placed on a paper sheet and scrutinized (from below) using a NdFeB commercial magnet, in order to extract magnetic particles from the powdered shell (see Video S1 in the Supplementary Materials). Then, the magnetic particles were placed on the VSM equipment following the same procedure as described above.

Furthermore, we used electron microscopy techniques to characterize the solid magnetic powder at the nanoscale. Scanning electron microscopy (SEM, FEI Quanta 3D coupled with a Bruker EDS) and transmission electron microscopy (TEM, FEI Tecnai G2-20 Super-Twin 200 kV, coupled with a Gatan Image Filter Quantum SE and an Oxford TEM Xplore EDS) analyses were performed at the Center of Microscopy of the Universidade Federal de Minas Gerais, Brazil. Selected area electron diffraction (SAD), energy-dispersive X-ray spectroscopy (EDS), and electron energy-loss spectroscopy (EELS) were used to assess the morphology, mineralogy, and composition of the magnetic powder. We also employed these techniques to analyze the cross-section of the *Limnoperna fortunei* shell in order to localize the magnetic nanoparticles. Finally, we used Raman spectroscopy to analyze both the magnetic powder and the shell cross-section.

Finally, we performed Raman analysis (LabRam-HR 800, Horiba Jobin Yvon, Japan). The spectrometer was equipped with a He–Ne laser (excitation at 632.8 nm) and an Olympus BX41 microscope with $10\times$ and $100\times$ LWD lenses. The laser was focused on a 1–2 $\mu m^2$ area of the sample (with the $100\times$ lens). The scattered light was collected by a monochromator and detected using a liquid nitrogen-cooled CCD. The spectrum scans were from 100 to 4200 $cm^{-1}$ with a 1.1 $cm^{-1}$ pitch. For most of our experiment, we scanned an 80–1250 $cm^{-1}$ window, with an acquisition time of 30 s, acquired 10 times to increase the signal-to-noise ratio. To avoid local heating in the sample—which can result in thermal transformation—the power of the laser was previously tested.

### 3. Results

#### 3.1. Testing Dissolution of Magnetite Particles in 1 M HCl Solution

A careful test was conducted to verify whether the 1 M HCl solution used for the demineralization of the bivalve shell could dissolve the iron present in the shell and re-precipitate it as magnetite nanoparticles. This test was crucial to determine whether the magnetic particles could have formed during the HCl shell demineralization experiment. However, after 24 h in 1 M HCl, the same initial weight (100 mg) of a stand-alone and known magnetite test sample was recorded, indicating that no magnetite mass dissolution occurred in this period. The same counterproof test was carried out with a 100 mg synthetic nanosized (30 nm) known magnetite sample. In this second test, nearly all of the magnetite nanoparticles had been dissolved after 24 h in 1 M HCl. The final weight was measured as 1.5 g.

#### 3.2. Direct Obtention of Magnetic Particles without Demineralization

Due to the possible influence of the biomineralization process on the occurrence of magnetic nanoparticles, we decided to simply grind the shells with commercial sandpaper (SiC) and verify the occurrence of magnetic particles.

We attempted and succeeded in directly obtaining magnetic particles without demineralization. After sanding the *L. fortunei* shells with commercial sandpaper, we were able to collect magnetic samples from the powder with the aid of a NdFeB magnet (Supplementary Materials, Video S1: Sanded *L. fortunei* mussel shell: collecting magnetic particles).

### 3.3. Chemical Shell Composition of Limnoperna fortunei Using ICP-OES and EDX

Table 1 shows the concentrations of different metals present in the freshwater *L. fortunei* mussel shells. Obviously, the concentration of calcium was very high, considering that it is the major component of the shell. Sr, Al, Fe, Mn, Mg, and Na were the only metals that achieved concentrations at or above a fraction of a percent. These metals and other trace heavier metals were clearly in the freshwater and assimilated at or beyond the gills. The highest was Na, which achieved a decimal percentage, likely because sodium is essential in all metabolic processes. The other metal element concentrations were very close to values obtained elsewhere [18], especially for calcium ions, as $CaCO_3$ constitutes more than 90% of the shell's composition. The iron concentration in the *P. perna* shell was close to those obtained in other studies of marine bivalves [19,20]; however, the iron concentration in the *L. fortunei* shell differed from that of other freshwater bivalves [20]

**Table 1.** Metal element composition of the shells of *Limnoperna fortunei* and *Perna perna* derived using ICP-OES technique (in µg/g and %).

| Sample | Sample Weight | Year | Al | Ba | Be | Ca | Cd | Co | Cr | Cu | Fe | K | Li |
|---|---|---|---|---|---|---|---|---|---|---|---|---|---|
| | (g) | | | µg/g | | % | | | | µg/g | | | |
| *Limnoperna fortunei* Shell | 0.5 | 2016 | 51.0 | 159.05 | <0.25 | 35.08 | <0.5 | <1 | 2.37 | 5.2 | 395.0 | 31.8 | <0.2 |
| *Limnoperna fortunei* Shell | - | 2022 | 41.1 | 161.1 | <0.2 | 36.07 | <0.5 | <1 | <4 | 1.2 | 105.9 | 36.7 | <0.5 |
| *Perna perna* Shell | - | 2022 | 21.0 | 3.3 | <0.2 | 38.64 | <0.5 | <1 | <4 | <0.5 | 19.8 | 133.1 | 1.3 |

| | Sample Weight | Year | Mg | Mn | Na | Ni | Pb | Sn | Sr | Ti | V | Zn | Hg |
|---|---|---|---|---|---|---|---|---|---|---|---|---|---|
| | (g) | | | | | | | µg/g | | | | | |
| *Limnoperna fortunei* Shell | 0.5 | 2016 | 235 | 180 | 2200 | 2.86 | 23.05 | <15.0 | 585 | 22.8 | 2.81 | 90.4 | <0.01 |
| *Limnoperna fortunei* Shell | - | 2022 | 180.2 | 30.9 | 5371.5 | <4 | <10 | <30 | 1200.9 | 3.2 | <1 | 2.7 | <0.02 |
| *Perna perna* Shell | - | 2022 | 159.0 | 12.6 | 6630.2 | <4 | <10 | <30 | 1054.6 | <1 | <1 | <2 | <0.02 |

The EDX results are presented in the Supplementary Materials (Table S1) and were in line with the results obtained by ICP-OES. This indicates that the iron concentration in *L. fortunei* shells was higher than that in *P. perna* shells.

### 3.4. UV–Vis Spectroscopy

Figure 3 presents the UV spectral absorption results for the magnetic particles extracted from the *L. fortunei* shell in Na-EDTA solutions and demineralized shell liquor in EDTA and Na–EDTA. NaFe(III) EDTA only, Na–EDTA only, and pure EDTA solutions are presented for comparison. It can be seen that EDTA did not absorb in the 190–450 nm region, while the solution with the demineralized *L. fortunei* shell in EDTA solution (see solution preparation in Methods) showed absorption in the 250–260 nm, indicating the formation of the EDTA–Fe complex. The UV absorption of the magnetic particles in both the Na–EDTA and NaFe(III)EDTA solutions peaked at 257 nm. At the lowest pH range, higher dissolution rates were realized as protons reached the activated state for the breaking of the residual O–Fe bonds [21].

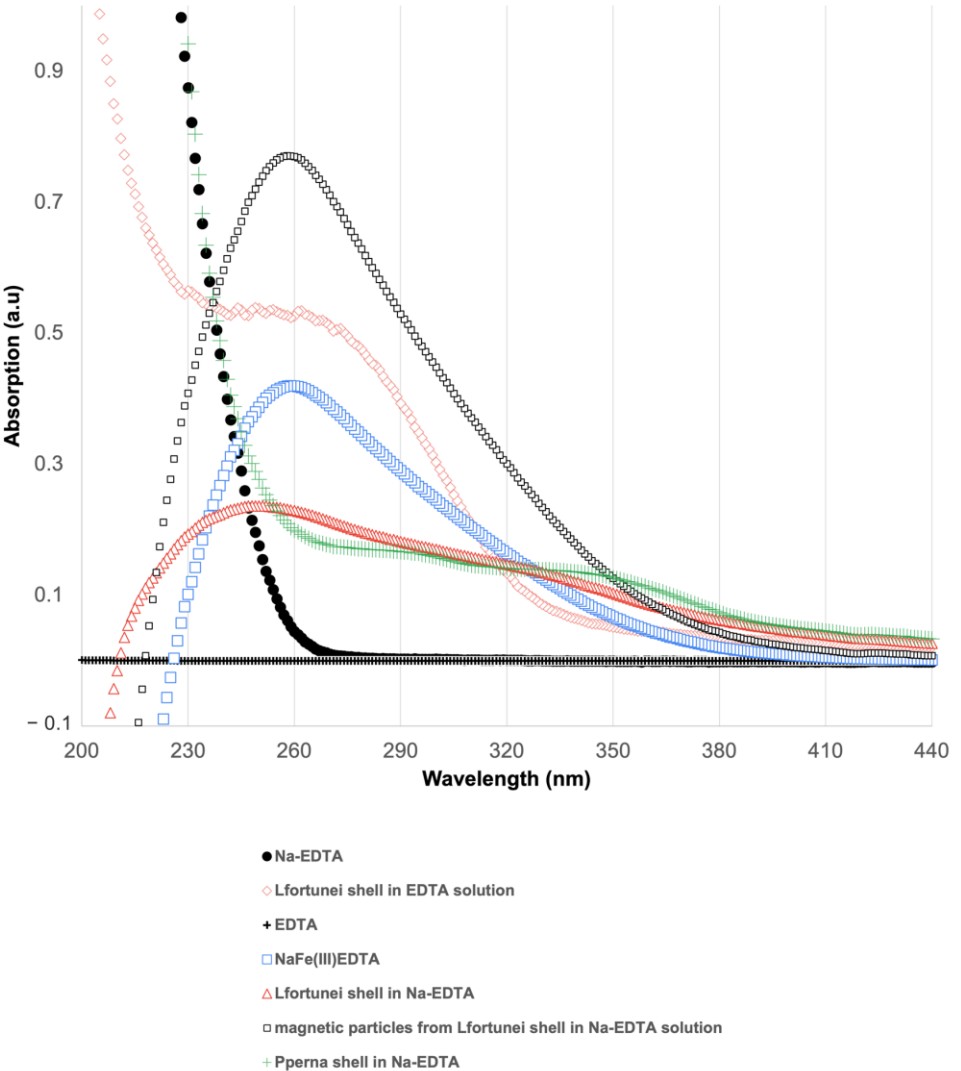

**Figure 3.** UV–Vis spectra of ethylenediaminetetraacetic acid (EDTA) (dash), Na–EDTA (circle), NaFe(III)EDTA (square) in solution, golden mussel (*Limnoperna fortunei*) shell powder in EDTA (diamond) and in Na–EDTA solution (triangle), and magnetic particles extracted from *L. fortunei* shell in Na–EDTA. Magnetic particles in Na–EDTA and NaFe(III)EDTA present absorptions at 257 nm; EDTA does not absorb in the UV part of the electromagnetic spectrum, while demineralized *L. fortunei* shell in Na–EDTA presents an absorption peak in the region of 230–260 nm.

*3.5. X-ray Diffraction*

Figure 4a,b present the X-ray diffractograms of the *L. fortunei* and *P. perna* mussel shells, respectively. The main XRD peaks of aragonite and calcite found in the shells were present, in accordance with the diffraction data for these two phases [22,23]. Aragonite was the main phase present in the *L. fortunei* shells. As for the *P. perna* shells, it was the only phase detected by XRD. Table 2 shows the concentrations of the aragonite and calcite phases, indicating that the *L. fortunei* shell was nearly all composed of aragonite, with the calcite appearing as a very thin layer attached to the periostracum [24]. The presence of the iron phase was not detected by XRD, as these phases occurred in concentrations lower than the limit of detection.

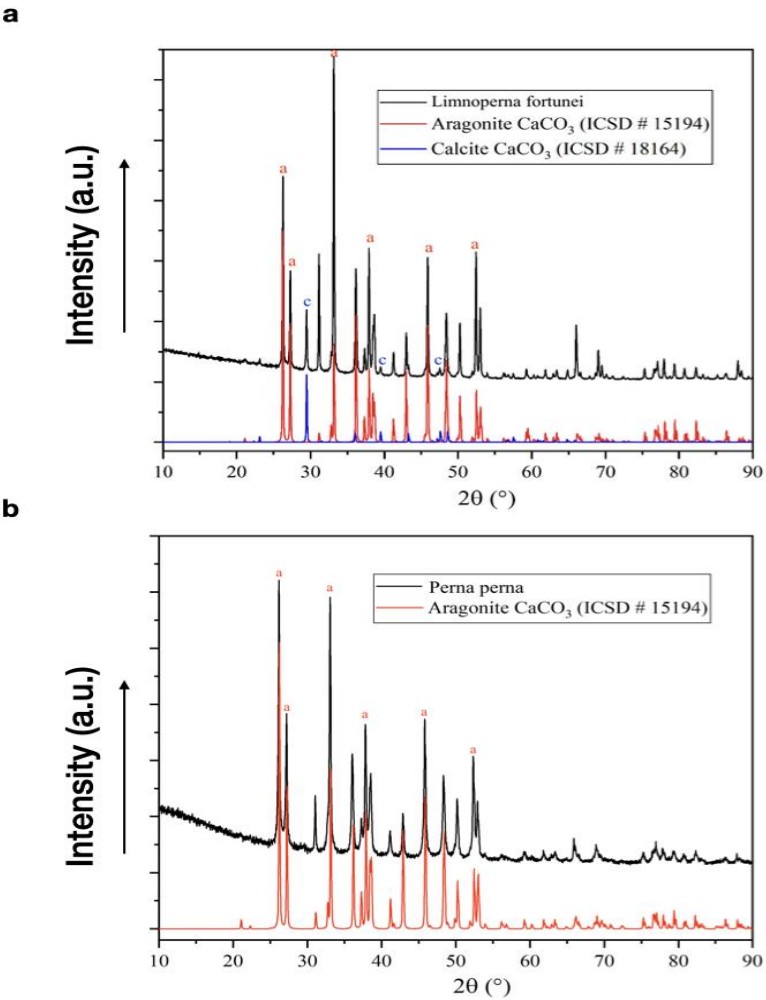

**Figure 4.** X-Ray diffractograms of mussel shells: (**a**) aragonite and calcite phases observed in the diffractogram of *L. fortunei*; and (**b**) aragonite was the only phase detected in *P. perna* using XRD technique. No iron phase was detected, due to the limits of detection. a: aragonite, c: calcite.

**Table 2.** Phases, concentration (%), composition, and crystalline system on the thermodynamic phases present in the shells of the freshwater *L. fortunei* and marine *P. perna* mussels.

| Phase | Chemistry Formula | Space Group | Crystalline System | Concentration (%) |
|---|---|---|---|---|
| Freshwater *Limnoperna fortunei* (golden mussel) | | | | |
| Aragonite | $CaCO_3$ | Pmcn | Orthorhombic | 97.48 |
| Calcite | $CaCO_3$ | R3c | Trigonal | 2.52 |
| Marine *Perna perna* (brown mussel) | | | | |
| Aragonite | $CaCO_3$ | Pmcn | Orthorhombic | >99.0 |

*3.6. Electron Microscopy*

Figures 5–9 show SEM and TEM images of the magnetic powder obtained from the demineralization of *L. fortunei* shells. The powder was attached to the magnetic stirring bar during the final demineralization step. The SEM data show the presence of calcium carbonate ($CaCO_3$) along with iron (hydr)oxide nanoparticles (Figure 5).

These mixed phases were not separated by sonication, as observed by TEM. Higher spatial resolution analysis of the magnetic powder was achieved by analytical TEM. The TEM data indicated either calcite or aragonite superimposed with the iron (hydr)oxides (i.e., magnetite, hematite, and goethite; Figures 6 and 7).

These calcium carbonates and iron (hydr)oxide phases were confirmed by EELS/EDS (Figure 6d,e) and SAD (Figure 7b,c). In Figure 6c, the EDS spectra at points 4 and 7 indicate the presence of sole CaCO$_3$. At points 1, 2, 4, and 5 (see Figure 6c), the EDS spectra indicate the occurrence of iron compounds. EELS analysis at the Fe $L_{3,2}$-edges of spectra taken at these points was performed to confirm the presence of magnetite. The white line ratios [Intensity (Fe-$L_3$): Intensity (Fe-$L_2$)] at points 1 (5.0), 2 (5.3), and 4 (5.4) agree with data reported for magnetite (5.2 $\pm$ 0.3) in the literature [25]. Figure 8 shows the TEM image, SAD pattern, and EEL spectrum of a sole magnetite nanoparticle found in the magnetic powder. Again, the magnetite and other iron oxide phases were mixed with either aragonite or calcite in the magnetic powder sample. We attempted to localize the magnetic nanoparticles in the *L. fortunei* shell cross-section, but it was unsuccessful. We did not find any iron oxide particles in the periostracum layer or the calcite crystals just after the periostracum.

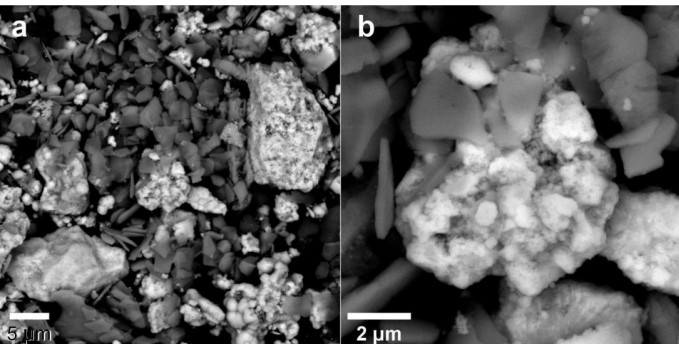

**Figure 5.** (**a**,**b**) Backscattering electron SEM images of the magnetic powder obtained from the demineralization of *L. fortunei* shells. The images show CaCO$_3$ plates (grayish) along with iron (hydr)oxides (brighter particles). The iron (hydr)oxides appear as agglomerates of nanoparticles entangled with the CaCO$_3$ plates, as shown in image (**b**).

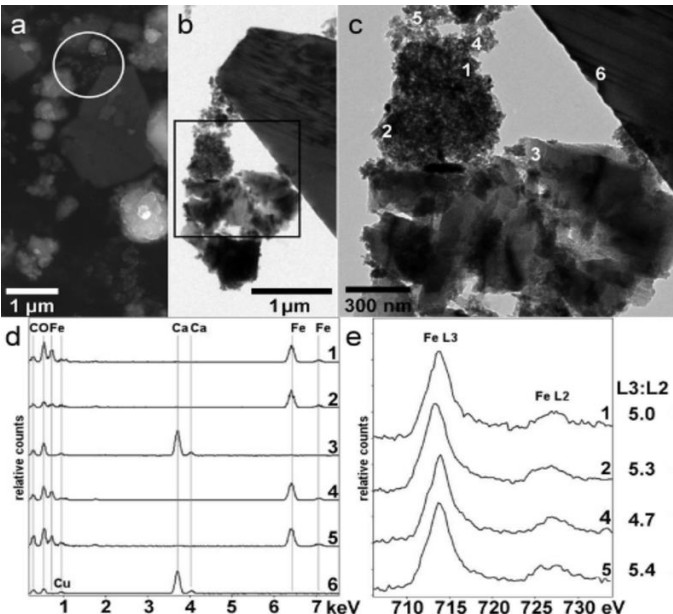

**Figure 6.** (**a**) Secondary electron SEM image and (**b**,**c**) bright-field TEM images of the magnetic powder obtained from the demineralization of the *L. fortunei* shell. (**d**) EDS spectra and (**e**) EEL spectra at the Fe *L*-edges taken from the points marked in image (**c**). The *L*3:*L*2 values are the ratios between the intensities of the Fe-*L*3 and Fe-*L*2 edges after background removal.

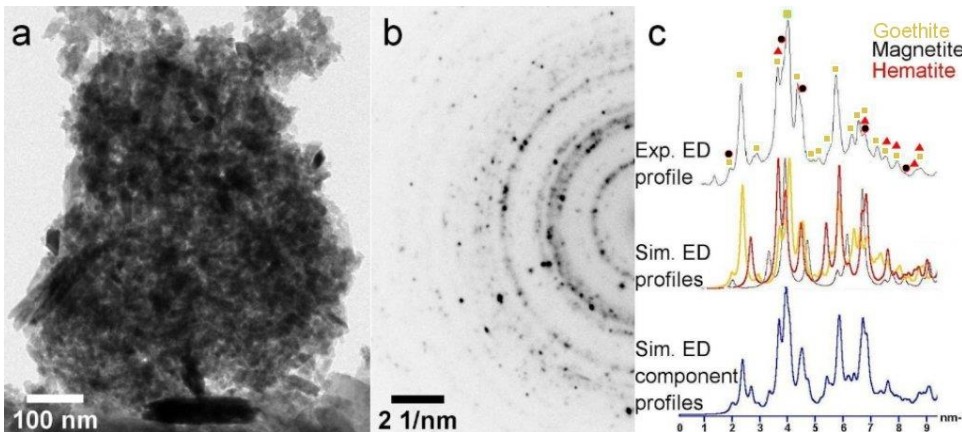

**Figure 7.** (**a**) Bright-field TEM image of an agglomerate of iron (hydr)oxide nanoparticles in the magnetic powder obtained from demineralization of *L. fortunei* shells. (**b**) SAD pattern (with inverse contrast) of the agglomerate of particles shown in image (**a**). (**c**) At the top, the experimental (Exp) electron diffraction (ED) profile of the SAD pattern shown in image (**b**). In the middle, the simulated (Sim) ED profiles of goethite (green), magnetite (black), and hematite (red). At the bottom, the simulated ED component profiles, considering equal contents of goethite, magnetite, and hematite. The simulated ED profiles were obtained using the JEMS© software (v. 3.4922U2010).

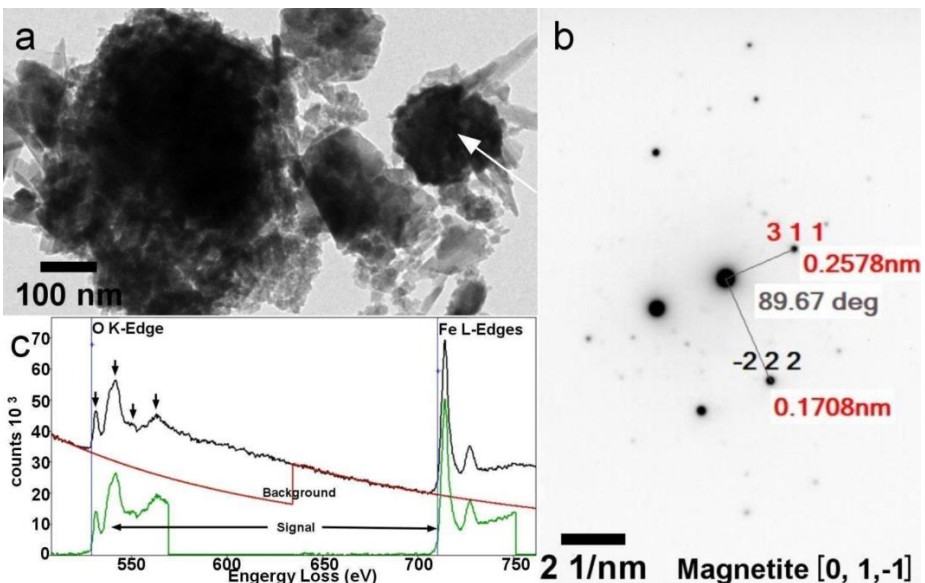

**Figure 8.** (**a**) Bright-field TEM image of an agglomerate of iron (hydr)oxide nanoparticles in the magnetic powder obtained from the demineralization of the *L. fortunei* shell. (**b**) SAD pattern (with inverse contrast) of the magnetite particle indicated by the arrow in image (**a**). (**c**) EEL spectrum at the O *K*-edge and Fe *L*-edges of the magnetite nanoparticle indicated in image (**a**). All four characteristic peaks at the oxygen *K*-edge for magnetite are indicated by black arrows.

Nevertheless, some roughly round nanoparticle aggregates of iron (hydr)oxide (likely goethite) were found in the nacre layer (Figure 9), associated with aragonite. Figure 9 shows SEM and TEM images of a section of the nacre layer with an iron-rich nanoparticle attached to it. The TEM—and especially HRTEM (Figure 9c)—results indicated that the round nanoparticle aggregates were made up of smaller (nearly 5 nm) nanoparticles. Fast Fourier transform (FFT) analysis of the HRTEM image confirmed that the nanoparticles were goethite.

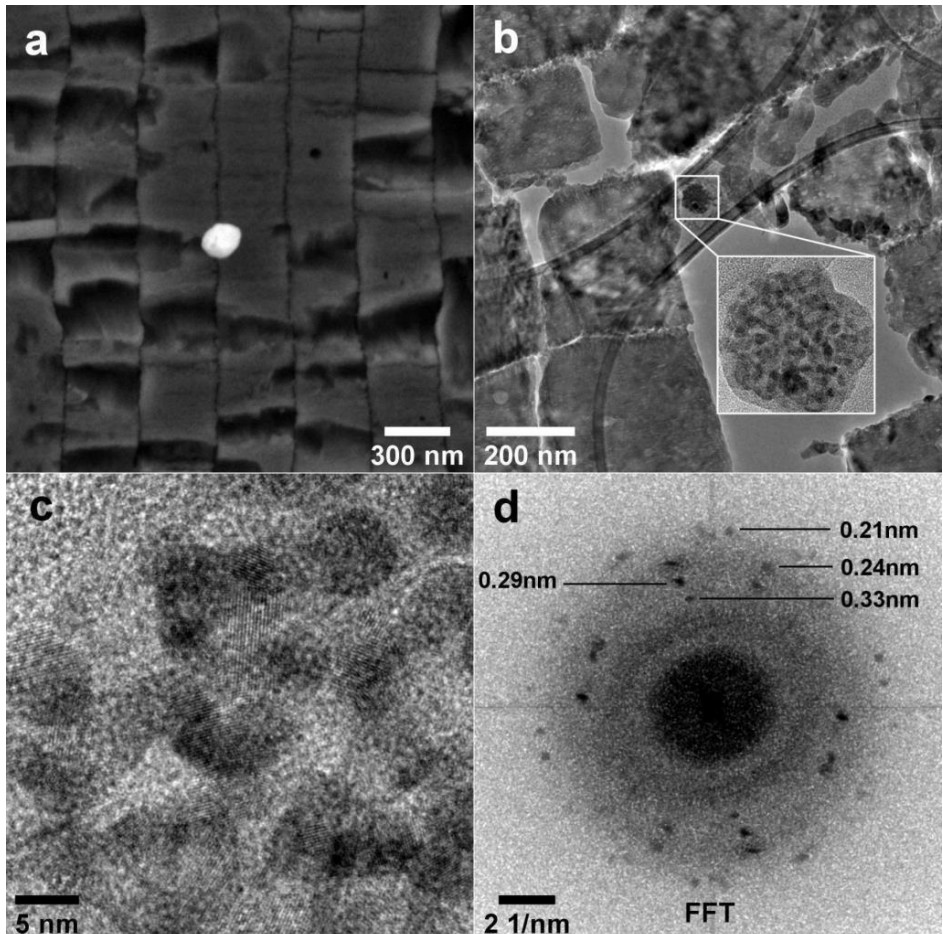

**Figure 9.** SEM and TEM images of the cross-section of the nacre layer of the *L. fortunei* shell. (**a**) Backscattering electron SEM image showing a round iron-rich nanoparticle attached to arag-onite tiles. (**b**) Bright-field TEM image of an ultra-thin cross-section layer (100 nm thick) showing a round nanoparticle aggregate between the aragonite tiles. (**c**) HRTEM image showing the crystalline nanoparticles of the aggregate marked in (**b**) in detail. (**d**) Fast Fourier Transform of the HRTEM image. The distances between the lattice fringes are indicated, which agree with goethite.

*3.7. Raman Spectroscopy*

The magnetic powder collected from demineralized shells was analyzed by Raman spectroscopy. Some of the Raman spectra are shown in Figure 10. Magnetite, goethite, and other iron (hydr)oxides were found. Spectrum 1 was acquired at a dark point seen in the light microscopy image, which shows the 666 cm$^{-1}$ line characteristic of magnetite. We found goethite in a yellowish spot of the sample observed at the microscope level (Figure 10, spectrum 2). Mixed phases were also observed at some points, as can be seen in spectrum 3, indicating the presence of both magnetite and goethite. Aragonite (Figure 10, spectrum 4) was found in the magnetic powder samples. These results corroborate the findings of the SEM and TEM analyses. During the Raman measurements, we observed the formation of hematite under the laser exposure at some points; therefore, the laser intensity was adjusted to avoid this phenomenon.

Additionally, Raman spectroscopy was employed to confirm the occurrence of mag-netite particles collected from mussel shells of *P. perna* (Figure 11a) and *L. fortunei* (Figure 11b) ground using a porcelain mortar and pestle.

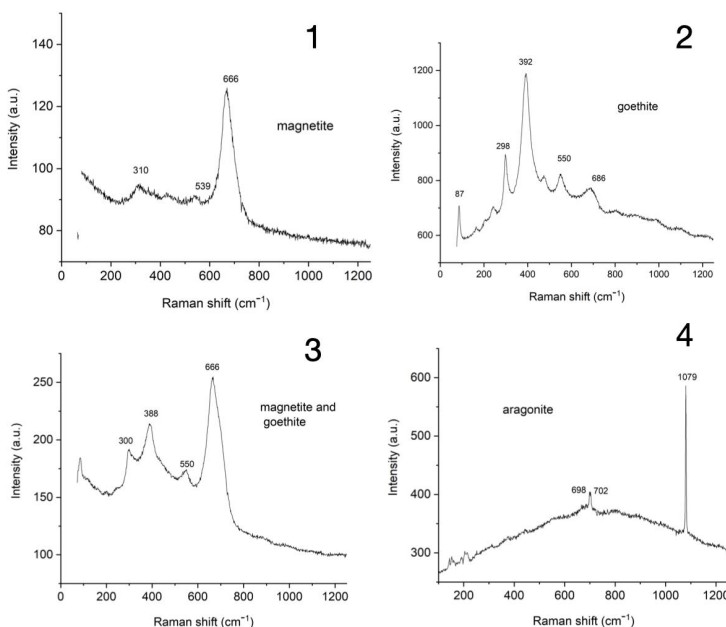

**Figure 10.** Raman spectra (1, 2, 3, and 4) of the magnetic powder obtained from demineralized *Limnoperna fortunei* shells.

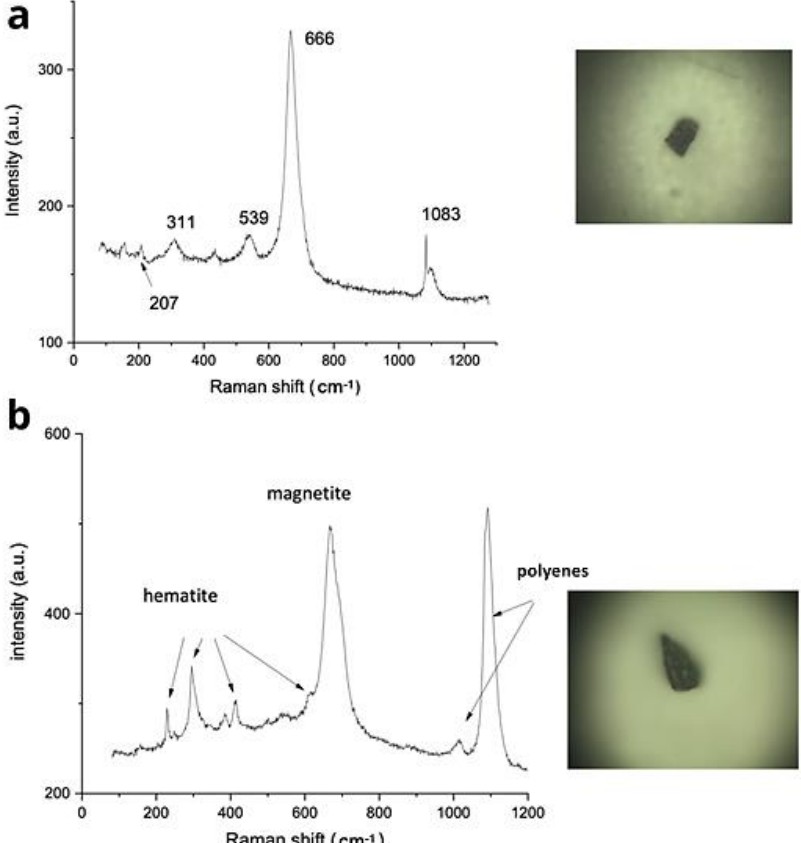

**Figure 11.** (**a**) Raman spectra of the magnetic particles obtained from *P. perna* shells. The spectrum shows magnetite (Fe$_3$O$_4$) lines at 666, 540, and 311 cm$^{-1}$, and aragonite (CaCO$_3$) lines at 1083, 207, and 155 cm$^{-1}$. The particle size is ~100 × 150 μm (image above right, 10× Objective); (**b**) Raman spectra of the magnetic particles obtained from *L. fortunei* shells. The spectrum shows magnetite, hematite, and polyene peaks. The particle size is ~25 × 50 μm (image above right, 50× Objective). Powder obtained from fresh shell, ground in a porcelain pestle and mortar.

Sections of gills from freshwater *L. fortunei* were also analyzed using Raman spectroscopy, in the search for a variety of micrometer-size particles of hematite, lepidocrocite, aragonite, calcite, and anatase (Figure 12). In other soft parts of the body of freshwater *L. fortunei*, we observed the presence of micrometer-size particles of hematite. There is a possibility that these larger particles, likely captured by the gills, might arrive inside the circulatory system of the mussel, as observed on many occasions using Raman spectroscopy.

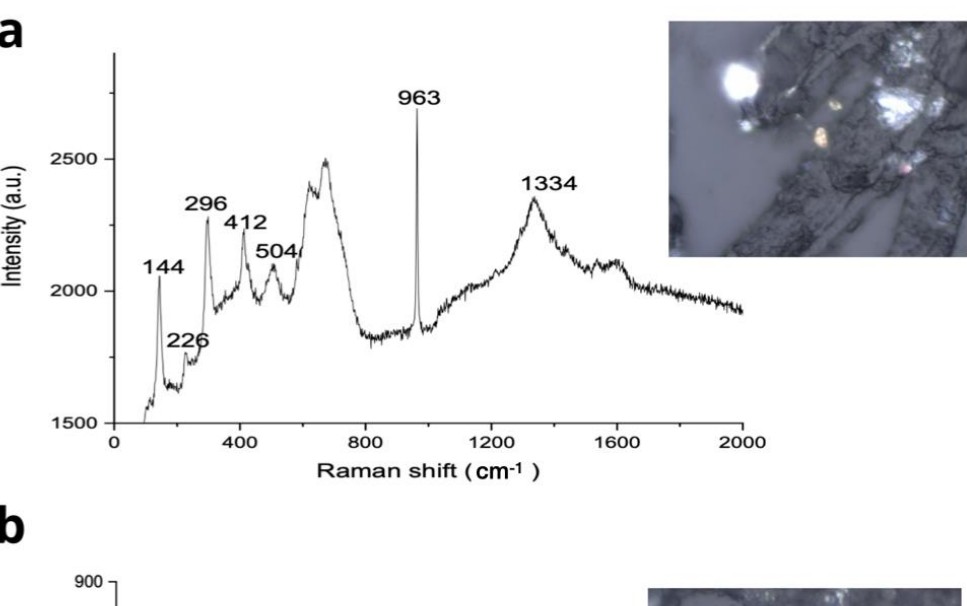

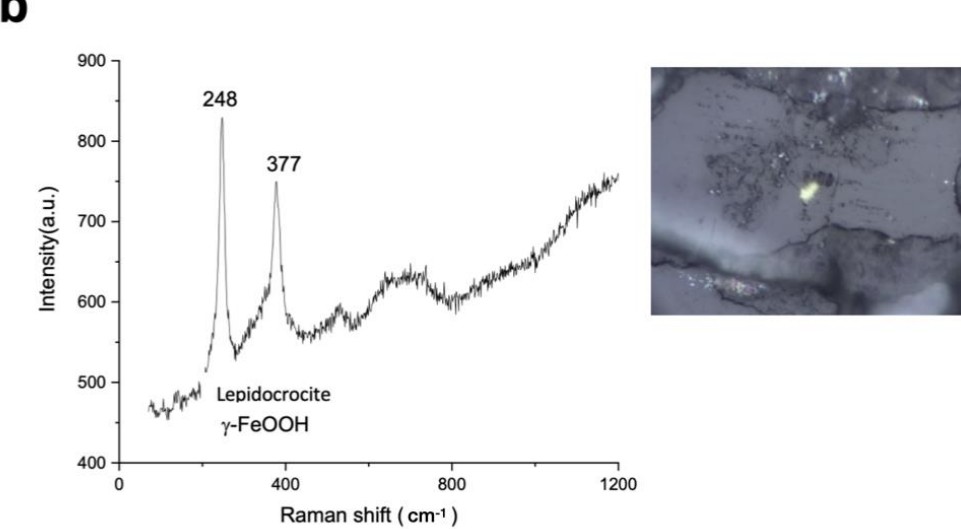

**Figure 12.** (**a**) Raman spectrum of a section of *L. fortunei* gills. Spectrum presents lines at 144 cm$^{-1}$ (anatase TiO$_2$), as well as 226, 296, 412, and 610 cm$^{-1}$ ($\alpha$-Fe$_2$O$_3$, hematite). The 963 cm$^{-1}$ line may indicate apatite, a calcium phosphate mineral; (**b**) spectrum with lines at 248 cm$^{-1}$ and 377 cm$^{-1}$ for ($\gamma$−FeOOH, lepidocrocite). Laser at 632.8 nm, CCD cooled to −126 °C, 100× lens objective.

### 3.8. Vibrating Sample Magnetometer (VSM)

To measure the magnetic properties of the *P. perna* and *L. fortunei* shells, we performed magnetometry using a vibrating sample magnetometer (VSM). In the *P. perna* shell fragments, we observed the predominant signal of diamagnetic materials from amorphous calcium carbonates, as seen by SEM/TEM. Although the magnetic signals observed in Figure 13(s1,s6,s7,s8,s9) were relatively low, a small magnetic hysteresis can be observed, indicating the presence of magnetic material in the mussel shells; for example, magnetite (Fe$_3$O$_4$) has ferrimagnetic order. The other fragments shown in Figure 13(s2–s5), on the other hand, presented a practically diamagnetic signal.

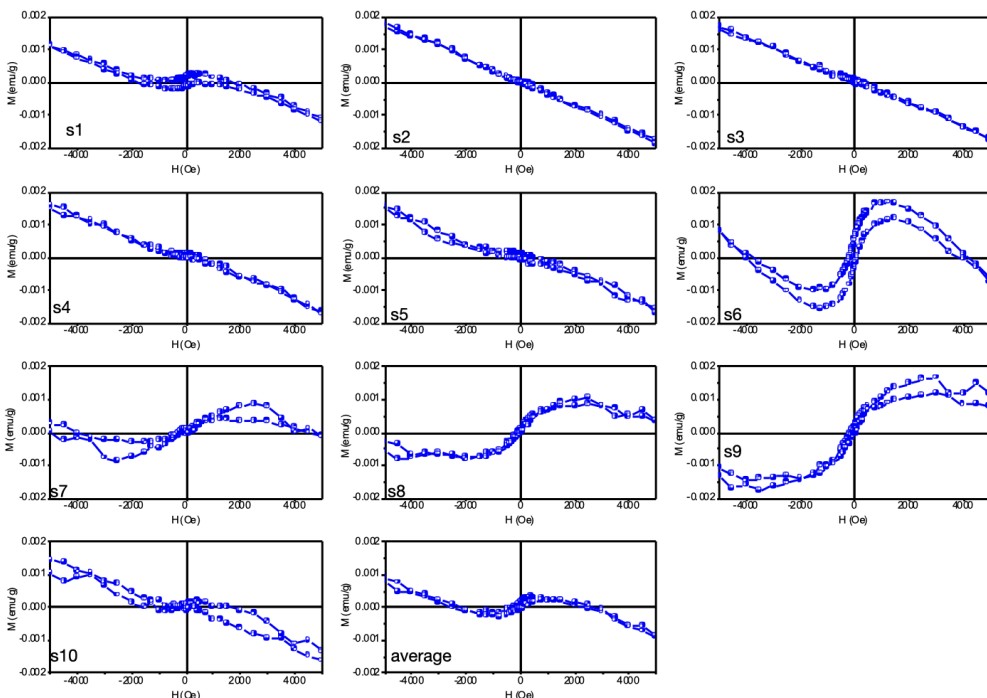

**Figure 13.** Magnetization M (emu $g^{-1}$) versus magnetic field strength H (Oe) measured in 10 pieces of the shell of the marine mussel *P. perna (*brown mussel). Pieces numbered (**s1**–**s10**) and the average of the assays is presented in the last graph (bottom, middle).

The magnetic curves that presented the highest concentration of magnetite still presented low coercivity and magnetic remanence, highlighting the fact that these magnetic materials present in the shells had low concentration and presented a superparamagnetic behavior characteristic of materials at the nanometric scale.

However, as magnetite (ferrimagnetic), hematite (antiferromagnetic), and/or goethite (antiferromagnetic) nanoparticles can be present in these fragments and competing, in magnetic property, with each other, the one with the highest intensity prevailed due to the volumetric measurement of the magnetometer VSM [26].

For the *L. fortunei* shell VSM test, we divided the shell into fragments and numbered them from 1 to 9, as shown in Figure 14. Again, we observed that some of the samples presented a ferrimagnetic characteristic (FI) for Figure 14 (specifically Figure 14(s1,s6,s9)), in addition to other components: diamagnetic (DM) present in Figure 14(s2,s4,s5); and paramagnetic (PM), present in Figure 14(s3–s7).

It is worth noting that the samples had low magnetization due to their low iron concentration: the PM, DM, and FI signals overlap in this concentration range. Pure iron material has about 221 emu $g^{-1}$ [27], while magnetite nanoparticles obtained through chemical/physical processes have an average value of 80 emu $g^{-1}$ [28,29]. The saturation magnetization, from lower to higher signal (FI), can be estimated to have had a concentration of 10–150 ppm of magnetite in the mass of these fragments analyzed.

Considering that the sensitivity of the VSM-7400 equipment is around $1 \times 10^{-7}$ emu and that the technique is volumetric, accounting for all magnetic signals present in the samples or their absence, some of these fragments may have included magnetic nanoparticles, but the total signal was masked by that from other non-magnetic components.

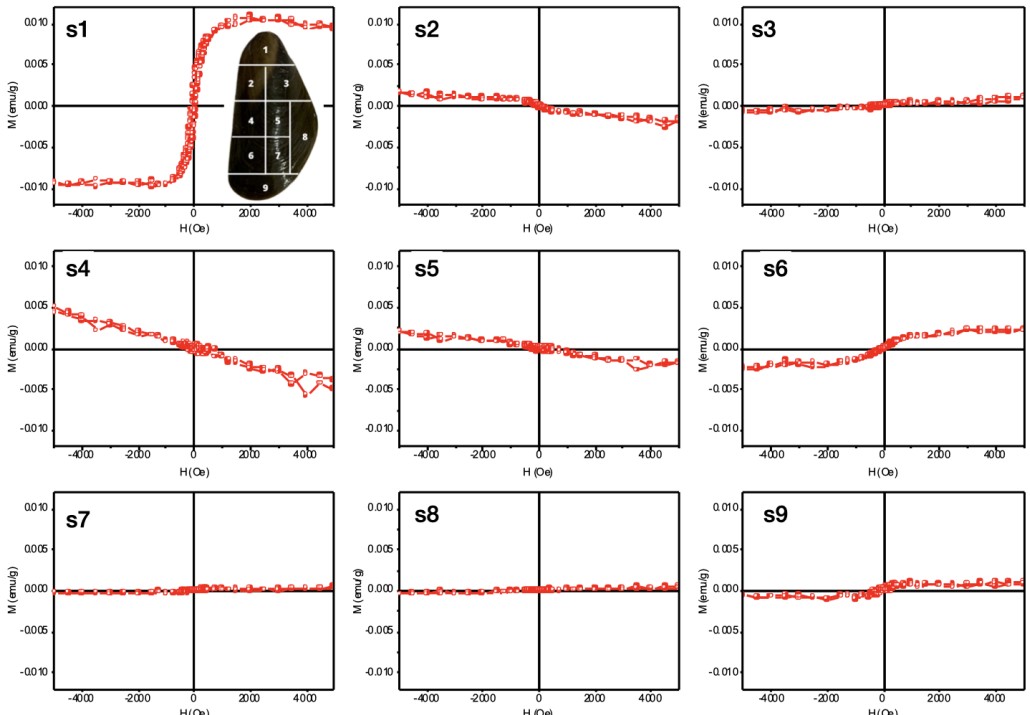

**Figure 14.** Magnetization M (emu/g) versus magnetic field strength H (Oe) measured in 9 (nine) pieces of the shell of freshwater mussel *L. fortunei* (golden mussel). Pieces were numbered (**s1**–**s9**) (see image in **s1**, top right).

## 4. Discussion

Going back to our questions, are the nanomagnetic particles present in the shells of either *L. fortunei* and *P. perna* simply ingested particles or those absorbed by the gills? Mussels are suspension feeders that pump and flow a huge volume of water each day through their bodies; as such, the magnetite nanoparticles could come from feeding on microalgae [30] or present in the water being continuously pumped in and out. Then, they could diffuse into the hemolymph, eventually ending up in the shell itself. Magnetic iron-oxides are found in a multitude of environments worldwide, simply by natural occurrence as well as due to human-made environmental pollution.

Bivalves continuously pump suspended microparticles and nanoparticles of many different compositions and minerals, both in rivers and the sea, with the most abundant being silica and aluminosilicate clays. However, none of these appeared in noticeable amounts in the shell composition, as indicated by the ICP-OES (Table 1) data. Previous studies [31,32] have come to the same conclusion: the presence of nano- or micro-particles of different compositions does not mean that they will be incorporated into the shell. For this reason, the composition of the shell is quite exclusively formed of $CaCO_3$, chitosan, and proteins.

Raman spectroscopy (Figures 11 and 12) indicated many micrometer-scale particles in the body of *L. fortunei*, even magnetite particles; however, the amount of iron detected by ICP-OES in the shell was much lower than 1%, similar to other detected heavy metals. This is quite understandable, considering that any particle could only arrive at the extra-pallial surface after traversing the bivalve mantle. The bivalve mantle is a rather complex structure [33], which has an intricate mechanism for selecting the inorganic particles that will arrive at the extrapallial external surface. Therefore, many microparticles (and, surely, many more nanoparticles) diffuse and are present in the soft body of the bivalves, but these are not freely able to cross and arrive at the outer face of the mantle. SEM-EDS imaging of the mantle of *L. fortunei* presented a quite complex structure, full of calcium associated with phosphate [34].

The impacts of iron cations on the crystallization, nucleation, and growth of $CaCO_3$ thermodynamic phases have been detailed in numerous scientific works in many areas. One line of explanation is based on the cation radius [31,32]. Calcium has a Pauling ionic radius of 99 pm and, in calcite—an ionic crystal—the Ca atoms have a face-centered distribution and the structure (viewed as a rhombohedron) has triangular $CO_3$ groups lying at the centers of each edge of the rhombohedron. This results in layers of $CO_3$ groups lying normal to the c-axis of calcite, with layers of Ca atoms lying between them. Iron ions ($Fe^{+2}$ and $Fe^{+3}$) with radii less than that of the calcium ion would replace calcium, as iron carbonate (siderite) is isostructural with calcite. According to this line of thought, not only iron but many divalent metal cations smaller than calcium could substitute for $Ca^{+2}$ [35].

Studies on the impact of iron on the crystallization of $CaCO_3$ have immediate economic importance, as the $CaCO_3$ scale can occlude pipes in water treatment systems, desalination plants, mineral, and crude oil pipelines, among others. Contrary to the above suggestion (easy replacement of Ca by Fe ions), experimental studies [36] have shown that, in an aqueous phase and the presence of minute amounts of ferrous iron $Fe^{+2}$, calcite will not occur at all or only marginally. The calcite phase is the most stable form of crystalline calcium carbonate, which is very difficult to dissolve as soon as it nucleates and grows on the walls of pipelines. However, the presence of ferrous ions does not prevent the nucleation and growth of aragonite. The reason for this behavior is not yet clearly understood but has been observed ever since the problem caught the attention of scientists [36,37].

Molecular dynamics studies [38] have also demonstrated that the growth of calcite is inhibited in the presence of ferrous ions and other divalent metal ions smaller than calcium. The initial incorporation of ferrous ions at the growth step of calcite is energetically favorable (exothermic) but, as the incorporation continues, it produces mismatches that ultimately make the growth of the crystal energetically unfavorable (endothermic) and stops it. On the other hand, ferric ion $Fe^{+3}$ also influences $CaCO_3$ nucleation and growth. As $Fe^{+2}$ is oxidized to $Fe^{+3}$ (even in very small amounts), the calcium carbonate present tends to crystallize in the calcite phase, as many works have reported. One explanation for why this happens is that the electronegativity of iron (1.8) is much higher than that of calcium (1); therefore, if iron and calcium coexist, the crystallization of calcium is directly affected. Evidently, other variables might interfere—especially acidity or basicity.

It has recently been reported that the acidification of seawater not only tends to corrode bivalve shells, but also alters their shell structure, with the aragonite nacre layer thickening at the middle of the shell while being wiped out from the acidic borders [39]. Aragonite is more easily dissolved, especially in acidic waters. However, at the center of the shell, it is not in contact with the acidic water and its thickening increases the mechanical strength of the shell. Therefore, it might be reasonable to think that the magnetic particles we encountered in both maritime and freshwater bivalves are related to the aragonitic nacre layer, which is a matter for further studies.

Finally, the suggested ability of Fe ions to interfere with the crystallization of $CaCO_3$ is based on the reasoning that the concentration of Fe should be in the ppm (mg $L^{-1}$) range, as confirmed by the ICP assays presented in this paper. One hypothesis, discussed by A. Mount et al. [40], is that calcite nucleates and grows inside hemocytes (at least in oysters). For the iron nano-compounds (oxides or hydroxides) to work as nucleating agents, this could be easier and less costly (in energy terms) for the hemocytes if nucleation occurs inside the cell compartment. Another possibility is that granular hemocytes deliver amorphous calcium carbonate (ACC) and iron nanoparticle aggregates at the extrapallial surface. The pH defines the nucleation phase, as $Fe^{+2}$ will favor aragonite nucleation, while $Fe^{+3}$ closer to the external environment will favor calcite formation. Both possibilities help to understand how the occurrence of crystalline $CaCO_3$ probably occurs by combining organic matrix induction and the nucleating agent oxidation state, as occurs in the process of scale formation.

The list of animals with the proven presence of magnetic particles continues to rise. However, to date, the presence of magnetic nanoparticles in the shell of mollusks has not

been reported. The scientific articles published so far have focused on the diversity of uses for the $CaCO_3$ present in shells [4]. However, heat treatment of up to 1000 °C or higher [5] is typically performed to obtain the shell powder, oxidizing any nanoparticles into $Fe_2O_3$. In addition, magnetite appears to be present in human organs, including the human brain [41,42]. More recently, an external source of magnetite nanoparticles present in the human brain was presented by researchers as the result of urban pollution [43]. According to their study, the morphology differentiates the two types.

Our own findings provide evidence of the magnetic particles present in the freshwater *Limnoperna fortunei* and saltwater *Perna perna* mussel shells. The TEM results indicated magnetite and maghemite, along with other Fe-(hydr)oxide nanoparticle aggregates close to the micrometer-size plates of aragonite. The Raman spectroscopy results indicated the occurrence of magnetite, goethite, other iron compounds, and mineral microparticles, along with aragonite particles. The results provided here help us to understand why the occurrence of magnetite particles has not been noticed so far: being nanoparticles, they would easily oxidize to other iron oxides.

Considering 5 nm or 10 nm spherical iron-rich particles and considering the *Limnoperna fortunei* shell weight of 300 mg, we arrive at an astounding number of iron-rich nanoparticles. If they were all magnetite, they could only be detached due to the magnetic tendency to aggregate (See Figure 2). Therefore, it is still a matter of importance to clarify whether magnetite biomineralization is concurrent with ACC, as has previously been reported for some bacteria [44]. Could the biogenic formation of ACC somehow help in the nucleation and growth of magnetite, or is it the other way around?

As $Ca^{+2}$ and $Fe^{+2}$ ions are toxic to cells, the nucleation and growth of calcium carbonate and iron oxide are a matter of survival for the cells. However, many studies have suggested that ACC nanoparticles a few nanometers in size are readily available in the water [45,46] and are probably captured by the gills. The Raman spectroscopy results presented here did demonstrate a variety of mineral microparticles inside the soft body, which may also be the case for the magnetite nanoparticles, as the phase change from iron (oxy)hydroxides (ferrihydrite) to iron oxides has been reported in chiton teeth.

At present, there is experimental evidence for the presence of magnetite nanoparticles (MNPs) in bacteria, chitons, fish, and humans [47]. The so-called "magnetoreception"—that is, spatial orientation by terrestrial magnetism—is a subject of scientific investigation [48] and controversy. Some experts on the subject have suggested that the formation of magnetite nanoparticles may have occurred before the emergence of eukaryotic cells [49]. The results obtained here indicated that magnetic particles occur as very small particles, at the nanometer scale. These biomineralized magnetic nanoparticles presented approximately equal volume and shape and might be considered as a source of nanoparticles for the medicine and molecular biology fields [50–52].

Recent advances in the comprehension of the role of hemocytes in the biomineralization of the shell [53,54] have indicated that part or all of the calcium that arrives at the mantle arrives as calcium carbonate ($CaCO_3$), possibly [55] in the amorphous state (ACC). According to these findings, the $CaCO_3$ nucleates and grows inside a fraction of the motile hemocyte cells, and is then transported to the mantle.

The energetic status of these aggregates is very important to consider. Thermodynamically, ACC has an excess of energy and is metastable. These sub-micrometer size aggregates are then delivered to the extrapallial space, where they are added to the inner faces of the growing shell. There, as suggested, it changes to crystalline phases (calcite and aragonite). This transformation from amorphous to crystalline is exothermic and tends to promote volume contraction, as the crystalline phase occupies less volume than the amorphous phase.

The concomitant biomineralization of magnetite and calcium carbonate has been recently reported in magnetotactic bacteria [44], and the authors believe that the biomineralization of more than one mineral phase could be a widespread phenomenon [56]. While the carbonate is amorphous (ACC), the magnetite is crystalline. While the ACC particles

are at the micrometer scale, the magnetite particles are nanometer-sized. If this is the case, magnetite nanoparticles may be distributed throughout all shell layers.

## 5. Conclusions

The dynamics of biomineralization—that is, the steps and sequences of events involved in the construction of the shell—in bivalves remain unresolved, even bearing in mind the important results [14,57,58], gradually but continuously obtained every few years in recent decades. Through this work, we believe we have added new information toward the solution to this problem, having found and characterized magnetic nanoparticles in the shells of two bivalve species. Magnetite ($FeO.Fe_2O_3$) and/or maghemite ($\gamma$-$Fe_2O_3$) nanoparticles may play a role in the process of shell construction. To be sure that the magnetite/maghemite particles in the shell were not a peculiarity of the freshwater exotic bivalve *Limnoperna fortunei*, we also identified magnetic particles in the shell of the marine bivalve *Perna perna*.

The role of the magnetic particles and their location in the shell is still under investigation, but the TEM results presented here demonstrate that they are nanoparticles. Coupled with the VSM magnetometry results, we found a faint ferromagnetic signal throughout the shells, indicating that they are well-distributed and not concentrated. As such, these individual particles of nanometer size may be homogeneously distributed through the entire volume of the shell. If this turns out to be associated with the actual process, then—at least for these two species—shell construction might be subject to the effects of ferromagnetic forces at the nanometer scale.

Finally, the results presented here reinforce the observation that simply grinding shells with commercial sandpaper (SiC, silicon carbide grinding media) produced collectible magnetic particles, especially from the shell of *L. fortunei*. As such, anyone with a bivalve shell (maritime or freshwater) and sandpaper can replicate these results (see Supporting Video S1: sanded *L. fortunei* mussel shell: collecting magnetic particles. Cleaned and dried *Limnoperna fortunei* shell was sanded with commercial sandpaper (SiC)).

**Supplementary Materials:** The following supporting information can be downloaded at: https://www.mdpi.com/article/10.3390/applnano4030011/s1, Table S1: Energy-dispersive X-ray fluorescence (EDX) results (in ppm) indicating the atomic elements present in the shell of the freshwater bivalve *Limnoperna fortunei* and in the marine bivalve *Perna perna*. Equipment: EDX 7000 Shimadzu. Video S1: Sanded *L. fortunei* mussel shell: collecting magnetic particles; Video S2: *P. perna* shell magnetic particles.

**Author Contributions:** Conceptualization and methodology, A.V.C.; investigation, A.V.C., C.C.S., M.S.D., C.S.M., E.T.F., A.K., V.M.R., G.U.N. and L.D.H.; writing—original draft preparation, A.V.C. and C.C.S.; writing—review and editing, A.V.C. and E.T.F.; visualization, A.V.C., C.C.S. and M.S.D.; project administration, A.V.C. All authors have read and agreed to the published version of the manuscript.

**Funding:** This research was funded by ANEEL—Brazilian National Agency for Electric Energy/CEMIG S.A., Companhia Energética de Minas Gerais, grant number PD-04951-0604/2017.

**Institutional Review Board Statement:** Not applicable.

**Informed Consent Statement:** Not applicable.

**Data Availability Statement:** Data supporting the reported results of this work are contained within the article and supplementary material. The pre-print is available at the platform bioRxiv—https://www.biorxiv.org//content/10.1101/2022.09.04.506556v1 accessed on 24 June 2023.

**Acknowledgments:** Our thanks go to CEMIG S.A.—Companhia Energética de Minas Gerais, Minas Gerais, Brazil, for the financial support (Project GT 604—control of the golden mussel: bioengineering and new materials for application in ecosystems and hydropower plants), BIOMINAS Brasil for helping on publication charges and to Ariete Righi of the Raman Spec. Lab., Phys. Dept., UFMG (Federal Univ. of Minas Gerais, Brazil).

**Conflicts of Interest:** The authors declare no conflict of interest. The funders had no role in the design of the study; in the collection, analyses, or interpretation of data; in the writing of the manuscript; or in the decision to publish the results.

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
