# Peer review of "Characterization of Magnetic Nanoparticles from the Shells of Freshwater Mussel L. fortunei and Marine Mussel P. perna"

_2673-3501, doi:10.3390/applnano4030011_

Round 1

Reviewer 1 Report

The present manuscript entitled “Characterization of magnetic nanoparticles from the shells of freshwater L fortunei and marine P perna mussel” by Cardoso et al., describes the magnetite (Fe3O4) nanoparticles were extracted from the shells of freshwater Limnoperna fortunei (Dunker 1857) and marine Perna perna (Linnaeus 1758). Furthermore, the TEM analysis indicated that the 50-100 nm round magnetic particles are in fact aggregates of 5-10 nm nanoparticles. The authors report an interesting work. The objective and justification of the work are clear. I was pleased to review your manuscript. The authors performed enough characterization techniques in the present study such as UV-Vis, XRD, Raman, SEM, TEM, and EDX etc.,. Therefore, I recommend it for publication. However, some issues are detailed below which need to be addressed before its final acceptance in Applied Nano.

I advise the authors to take the following points into account while revising their manuscript.

Comment 1: There are some typographical and grammatical errors in the manuscript text, so the authors need to correct them in the revised manuscript. Carefully check the scientific names in the manuscript must be in italics, also correct the subscript errors.  For e.g. section 2.12 subheading “2.12. TEM e SEM Experiments” should be “2.12. TEM and SEM Experiments”; Line 342, “190-450nm” should be “190-450 nm” Line 373, “diffractogram of L fortune should be diffractogram of L fortunei”; Line 86, ICP-OES and XRD appeared the first time in the manuscript, so include the full forms of ICP-OES and XRD;  Figure 1, X-axis title Wavelength(nm) should be Wavelength (nm); Figure 10, X-axis title Raman Shift (cm-1) should be Raman Shift (cm-1)

Comment 2: L fortune and Perna perna is written in italics in some places of the manuscript and in some places not in italics. So, I strongly suggest the authors cross-check throughout the manuscript and correct it to italics and maintain uniformity.

Comment 3: The whole manuscript must be cross-checked thoroughly for English editing, grammatical, spelling mistakes, and syntax errors. So, I suggest the author's English language should be polished.

Comment 4: The Abstract needs to be revised, let the author focus main points and explain the research question clearly, and also mention all performed characterization technique's names in the abstract section.

Comment 5: In XRD analysis, I suggest the authors include the major peak positions of L fortunei and discuss them with the supported literature.

Comment 6: Authors mentioned in the manuscript indicated that the 50-100 nm round particles are aggregates of 5-10 nm nanoparticles. So, I suggest the authors include the particle size distribution graph of Figure 9(c) using Image J software to support the statement.

Comment 7: authors mentioned EDX results in the table form in the supplementary file (Table S3), it is better to include also the original EDX spectrum of Limnoperna fortunei bivalve and P perna bivalve along with Table S3.

Comment 8: The conclusion part is too lengthy authors can update the conclusion in a short and precise format.

Comment 9: The homogeneity of the reference section needs to be maintained. In some references, subscript errors are there (For e.g. Reference 39) also some journal names are written in short form (For e.g. Reference 15). So please check and revise accordingly to the journal's instructions.

Moderate editing of English language required

Reviewer 2 Report

The article "Characterization of magnetic nanoparticles from the shells of 2 freshwater L fortunei and marine P perna mussel" by Cardoso et al is an experimental study of nanoparticles extracted from mussel shells. It is an interesting read even if you are not from the field, and the authors used a large variety of analytical methods.

I have 2 major concerns that would need to be addressed from my viewpoint, both of which are not related to the science and methodology.

1) The article reads more like a lab report (or a thesis report - perhaps it is the outcome of a thesis even) than a journal article, and the results cannot be clearly explained in the summary conclusion. Rather different scenarios are suggested for the role and origin of the magnetic nanoparticles. That in itself is not a problem, however the writing style of the entire manuscript needs a major overhaul. Phrases like ""..one of us noticed..." [p1l37] "..investigating the literature we noticed..." [p2l59] are not suitable for a journal type article - again this is lab book style.

Adding to this, the english language is simply not acceptable as is. Many sentences are incomprehensible or bear resemblence to the authors' native language [nacar - nacre in the abstract etc], suggesting the use of an auto translate software.

To addres both of the above issues, I recommend extensive editing with the help of a native english speaker.

The english language is simply not acceptable as is. Many sentences are incomprehensible or bear resemblence to the authors' native language [nacar - nacre in the abstract etc], suggesting the use of an auto translate software [I am not saying that AI was used, but it reads like it].

To addres both of the above issues, I recommend extensive editing with the help of a native english speaker.

Reviewer 3 Report

June 13, 2023

Review on the the paper applnano-2460473

 Title “

 Characterization of magnetic nanoparticles from the shells of 2 freshwater L fortunei and marine P perna mussel
Authors: Cardoso et al.

which was submitted for publication to Applied Nano.

The authors have carried out a thorough study of mussel shells by a large variety of techniques to characterize their chemical composition and the various crystalline phases present with special attention to the occurrence of magnetic nanoparticles in the shells. A detailed discussion is given about the possible origin of these magnetic nanoparticles and their possible formation mechanisms. Although the authors admittedly are not able to give definite answers to the questions raised by their studies, the large amount of valuable experimental results provided in the manuscript makes this work worth of publication in Applied Nano.

There are however some suggestions for improving the paper before final acceptance for publication. The authors should take into acount these comments and revise their paper accordingly.

Technical comments:

1.             Table 1: In the caption, “mg/g” is specified. On the other hand, in the table itself “ug/g”. One may guess that this latter wanted to be mikrogram/gram”. Pleace chekc which is correct. If microgram, then use “greek mu” instead of “u”. Please check the whole text.

2.             The captions to Fig. 13 and Fig. 14 are not completely clear. It appears as if the captions referred to things which can not be found in the figures. Please check carefully thes captions.

3.             Section 4 contains only one subsection, namely 4.1 which is rather strange. Either apply a subsection 4.2 or remove the heading of subsection 4.1.

Comments on English: The English is generally acceptable. However, the authors are requested to check their text carefully since some sentences are not sufficiently clearly formulated. This partly due to the fact that the verb and noun in some sentences are not always properly matched with each other concerning plural and singular forms. A careful check for more appropriate punctuation would also of great help. Some suggestions for further improvement are given below.

1.             Page 1, line 32: “Since then, the presence of biomineralized magnetite in animal species has been growing steadily [2,3].” Evidently, the “presence” does not grow at all. What can grow is the number of papers describing the presence of magnetite if I correctly understand your intended mening. Please revise the sentence accordingly (the same in the abstract).

2.             Page 1, line 35: “presence of magnetic on the shell”. Did you mean “magnetite in the shell”? (the same in line 76). Please check and correct. Also, you often use the term “on the shell”. Please check if “in the shell” would be the appropriate form according to the intended meaning (“on the shell” = “on the surface of the shell” whereas “in the shell” = “inside the shell”). Apparently, “in the shell” as written in the black textbox of Fig. 2b. Please check the whole manuscript.

3.             Page 1, line 35: “accidently” --> “accidentally”

4.             Section 2.4: Please note that “weight” is a noun only whereas the verb form is “weigh”. Check the whole text.

5.             Page 20, line 532: “we performed magnetometry Vibrating Sample Magnetometer (VSM)” --> “we performed magnetometry by vibrating sample magnetometer (VSM)”. Please do not capitalize only in the abbreviation VSM.

see report above
